# High-dose corticosteroid pulse therapy increases the survival rate in COVID-19 patients at risk of hyper-inflammatory response

**Miguel Ángel López Zúñiga**[1,2�]*, **Aida Moreno-Moral**[3�], **Ana Ocaña-Granados**[4], **Francisco Andrés Padilla-Moreno**[5], **Alba María Castillo-Fernández**[5], **Dionisio Guillamón-Fernández**[6], **Carolina Ramírez-Sánchez**[7], **María Sanchez-Palop**[8], **Justo Martínez-Colmenero**[2,5], **María Amparo Pimentel-Villar**[9], **Sara Blázquez-Roselló**[10], **José Juan Moreno-Sánchez**[5], **María López-Vílchez**[5], **Inmaculada Prior-Sánchez**[11], **Rosario Jódar-Moreno**[12], **Miguel Ángel López Ruz**[1]

1 Infectious Diseases Unit, Hospital Universitario Virgen de las Nieves, Granada, Spain, 2 Grupo de investigación CTS 990: GEAPACECP, Universidad de Jaén, Jaén, Spain, 3 Independent Scholar, Cambridge, United Kingdom, 4 Allergology Service, Complejo Hospitalario de Jaén, Jaén, Spain, 5 Internal Medicine Service, Complejo Hospitalario de Jaén, Jaén, Spain, 6 Otorhinolaryngology Service, Hospital Universitario San Cecilio, Granada, Spain, 7 Pathological Anatomy Service, Hospital Universitario Virgen de la Victoria, Málaga, Spain, 8 Pneumology Service Complejo Hospitalario de Jaén, Jaén, Spain, 9 Haematology Service, Complejo Hospitalario de Jaén, Jaén, Spain, 10 Nephrology Service, Complejo Hospitalario de Jaén, Jaén, Spain, 11 Endocrinology Service, Complejo Hospitalario de Jaén, Jaén, Spain, 12 Otorhinolaryngology Service, Complejo Hospitalario de Jaén, Jaén, Spain

☯ These authors contributed equally to this work.
* miguelangellopezzuniga@gmail.com

## Abstract

### Objective

Test whether high dose corticosteroid pulse therapy (HDCPT) with either methylprednisolone or dexamethasone is associated with increased survival in COVID-19 patients at risk of hyper-inflammatory response. Provide some initial diagnostic criteria using laboratory markers to stratify these patients.

### Methods

This is a prospective observational study, 318 met the inclusion criteria. 64 patients (20.1%) were treated with HDCPT by using at least 1.5mg/kg/24h of methylprednisolone or dexamethasone equivalent. A multivariate Cox regression (controlling for co-morbidities and other therapies) was carried out to determine whether HDCPT (among other interventions) was associated with decreased mortality. We also carried out a 30-day time course analysis of laboratory markers between survivors and non-survivors, to identify potential markers for patient stratification.

### Results

HDCPT showed a statistically significant decrease in mortality (HR = 0.087 [95% CI 0.021–0.36]; P < 0.001). 30-day time course analysis of laboratory marker tests showed marked

**Data Availability Statement:** All relevant data are within the manuscript and its Supporting information files.

**Funding:** The author(s) received no specific funding for this work.

**Competing interests:** The authors have declared that no competing interests exist.

differences in pro-inflammatory markers between survivors and non-survivors. As diagnostic criteria to define the patients at risk of developing a COVID-19 hyper-inflammatory response, we propose the following parameters (IL-6 > = 40 pg/ml, and/or two of the following: C-reactive protein > = 100 mg/L, D-dimer > = 1000 ng/ml, ferritin > = 500 ng/ml and lactate dehydrogenase > = 300 U/L).

## Conclusions

HDCPT can be an effective intervention to increase COVID-19 survival rates in patients at risk of developing a COVID-19 hyper-inflammatory response, laboratory marker tests can be used to stratify these patients who should be given HDCPT. This study is not a randomized clinical trial (RCT). Future RCTs should be carried out to confirm the efficacy of HDCPT to increase the survival rates of COVID-19.

## Introduction

COVID-19 is a disease caused by SARS-CoV-2, originally described in Wuhan, China, in December of 2019. It is postulated that 80% of the infected population experience no symptoms or mild symptoms, while 20% are hospitalized, with 5% requiring intensive care, with a 50% mortality rate in these cases [1–3]. The course of the disease has been divided into three phases: a first phase characterized by a viral infection in the respiratory tract; a secondary pulmonary phase characterized by lung infection with a non-hypoxic stage (phase IIA) and leads into a hypoxic stage (phase IIB); and a third hyper-inflammatory phase [4]. Clinical experience has shown that, according to the age of the subject, the distinct phases of COVID-19 can present more or less virulently: while the tolerance of the first virulent phase decreases with age, the last hyper-inflammatory phase can be life-threatening in younger patients.

The hyper-inflammatory phase has the highest mortality rate, due to what has been described as a "hyper-inflammatory response". This hyper-inflammatory response is characterized by overproduction of early response pro-inflammatory cytokines that can lead to multi-organ failure and death [5–7]. Due to the urgency of this pandemic, many interventions have been tried with the intention of counteracting this hyper-inflammatory response [8–10]. Some of these interventions include drugs blocking IL-6 (such as Tozilizumab [11, 12]), IL-1 (Anakinra) or corticosteroids at different doses. However, the use of the latter has proved controversial and continues to be a subject of debate.

In this study, we examine whether there is an association between high-dose corticosteroid pulse therapy (HDCPT) and a decreased risk of death in COVID-19 patients with high inflammation levels, alongside other interventions. In order to explore some of the diagnostic criteria that could be used to decide which patients may benefit the most from HDCPT, we also analyse differences in laboratory markers between survivors and non-survivors throughout the course of COVID-19.

## Methods

### Inclusion criteria

We recruited all of the patients that entered the Complejo Hospitalario de Jaén (Hospital of Jaén, Spain) with confirmed or suspected COVID-19 from the 4th of February 2020 to 30th of April 2020, and who were more than 18 years old. 318 met the inclusion criteria of SARS-CoV-2 detection by PCR or serology (n = 272, 71.2%) or with high clinical suspicion (n = 46, 16.9%), defined as having bilateral pulmonary infiltrate or lymphopenia with a

concordant clinical profile. All patients were of Western European descent. This study was approved by the Ethics Committee of the Complejo Hospitalario de Jaén (Hospital of Jaén), Spain (0946-N-20). According to the local ethics committee regulations, verbal consent was obtained from patients that joined the study, and recorded in each patient's medical record.

## Variables measured and study design

We carried out a prospective observational study where clinical data was collected from all patients meeting the inclusion criteria and retrospectively compared survivors and non-survivors. Upon arrival to the hospital we registered the following: age, sex, date of the start of the COVID-19 symptoms and the presence of: dyspnoea, cough, fever, asthenia, anosmia and the oxygen saturation levels. The following details of the medical history were also registered: hypertension, smoking history, chronic obstructive pulmonary disease (COPD), asthma, chronic heart disease (CHD), atrial fibrillation, diabetes mellitus and whether the patient was under any oral or inhaled corticosteroid therapy at the time of hospitalisation (no matter the duration), had a tumour or was immunodepressed (i.e. patients who were taking immunosuppressors, had human immunodeficiency virus (HIV), or were immunosuppressed due to long term therapy with oral or inhaled corticosteroids). In addition, we registered whether patients had been taking angiotensin-converting enzyme (ACE) inhibitors / angiotensin receptor blockers (ARBs). In each of these patients, SARS-CoV-2 PCR and/or serology (IgM and IgG) tests were carried out.

We collected the results of all the laboratory tests performed from the start of hospitalisation until the end point of either death or hospital discharge. All the patients who were in intensive care unit (ICU) had laboratory tests every 24h. Patients outside of ICU had laboratory tests every 48h unless showing worsening symptoms (in these cases they were tested every 24h). In these laboratory tests, the levels of forty five markers were measured including hemogram, glomerular filtration rate, creatinine kinase, triglycerides, lactate dehydrogenase, interleukin 6 (IL-6), ferritin, serology for HIV, immunoglobulins and vitamin D, international normalized ratio (INR), D-dimer, prothrombin time and partial thromboplastin time (see the full list in S1 Table). Upon arrival to the hospital, we also took a chest X-ray and computed a quick sepsis related organ failure assessment (qSOFA) [13].

During their hospital stay, we evaluated the need for: oxygen supplementation and the maximum oxygen flux required (we considered high oxygen requirements oxygen volumes higher than 10L/min); mechanical assisted ventilation (either invasive or non-invasive); and intensive care requirement. We also registered all the medications taken during their hospital stay: hydroxychloroquine, lopinavir/Ritonavir, immunoglobulin therapy, tozilizumab, anakinra, azytromycin, vitamin D supplementation, anticoagulation and corticosteroid therapy. For anticoagulation therapy, we used either low-molecular-weight-heparin (LMWH) or direct-acting oral anticoagulants (DOACs) at three different dosages: prophylactic 3,500–4000 IU/day; intermediate 5,000–6,000 IU/day; or full 115–150 IU/kg/day (in all cases this medication was kept throughout the whole hospital stay).

Within the corticosteroid therapy, we distinguished between high dose corticosteroid pulse and low dose corticosteroid therapy. High dose corticosteroid pulse therapy was defined as a daily dose of at least 1.5mg/kg/24h of methylprednisolone or dexamethasone equivalent. The standard high dose corticosteroid pulse therapy duration was 3 days. In some patients who did not improve after 3 days, the treatment was extended to 5 days. In two patients, the treatment was shortened to 2 days due to the significant recovery observed. High dose corticosteroid pulses were given to patients following the criteria previously suggested from empirical observations and the guidelines used for the macrophage activation syndrome [14]: either IL-6 of at

least 40 pg/ml and/or two of these: ferritin, triglycerides and D-dimer of at least 300 ng/ml, 300 mg/L and 1000 ng/ml respectively. HDCPT was administered right after the detection of these marker levels independently of whether the patient was in intensive care or not. Not all of the patients that met these criteria received high dose corticosteroids pulses: out of 158 patients, 48 received high dose corticosteroid pulses. There were also 16 patients that received high dose corticosteroids pulses due to their critical clinical status, even though they did not meet these high-inflammation criteria. These 16 patients received high dose corticosteroids pulse therapy because they developed severe respiratory failure and did not respond to the standard COVID-19 clinical practice at that time, including pharmacological (hydroxychloro-quine, azithromycin and lopinavir-ritonavir) and physical interventions (i.e. postural changes). Low dose corticosteroid therapy defined as lower than 1.5mg/kg/24h of methylpred-nisolone or dexamethasone equivalent and it was administered to patients who had had a bronchospasm, following standard clinical guidelines.

## Statistical analysis

To test associations between outcome and demographics and clinical variables at entry, Student t-tests or Fisher tests were performed for numerical and categorical variables respectively (Table 1).

To assess treatment effect, a multivariate cox regression model was fit to the entire cohort using the following covariates: age, sex, hypertension, chronic obstructive pulmonary disease, asthma, chronic heart disease, atrial fibrillation, obesity, tumour, ACE inhibitors / ARBs, whether the patient was taking corticosteroids at the time of hospitalisation, whether the patient was immunosuppressed, whether the patient was given high oxygen volumes (>10L), diabetes, qSOFA, hydroxychloroquine, Azithromycin, Lopinavir/ritonavir, interferon, low dose of corticosteroids, HDCPT, Tozilizumab, vitamin D supplementation and anticoagulation therapy (at either intermediate, full or prophylactic dose). Using this multivariate Cox model, hazard ratios (HRs) and 95% confidence intervals (CIs) were computed.

Differences in laboratory markers between survivors and non-survivors during the first month of disease were computed by fitting a temporal trend using a regression spline. Then, a moderate F-test on the time: survival/non-survival interaction parameter was carried out to assess significance between the two groups using the function *ns*, *lmFit* and *eBayes* from the R packages *splines* and *limma* [15]. P values were corrected for multiple testing and the false discovery rate, FDR was computed by using the Benjamini & Hochberg method (R function *p. adjust*). The significance level considered in all analyses was 0.05. All statistical analyses were carried out using R (version 3.6.0).

# Results

## Characteristics of the cohort

We recruited all adult patients entering the Complejo Hospitalario de Jaén (Hospital of Jaén, Spain) from the 4th of February 2020 to the 30th of April 2020 with high suspicion of COVID-19, totalling 318 patients, as outlined in the methods. According to the NIH guidelines [16], out of the 318 patients included in our study, 20 patients were moderate (6.3%). The rest of the patients were all severe or critical (n = 238, 93.4%). Unfortunately, we cannot distinguish between severe and critical patients, since we do not have data for respiratory or multi organ failure, and not all critical patients were admitted to intensive care due to overcrowding at the height of the pandemic. The patients entered the hospital on average 7.79 days after first showing COVID-19 symptoms. The mean age was 64.9 (SD 14.1), ranging from 19 to 96 years. 186 were men (58.5%) and 132 were women (41.5%). All patients were of Western European

**Table 1. Patient demographics and clinical characteristics.**

| | Total (n = 318) | Survivors (n = 271) | Non-survivors (n = 47) | P value |
|---|---|---|---|---|
| Age | 64.9 (14.1) | 63.3 (13.6) | 73.9 (13.7) | <0.001 |
| Sex | | | | |
| Women | 132 (41.5%) | 112 (41.3%) | 20 (42.6%) | 0.874 |
| Man | 186 (58.5%) | 159 (58.7%) | 27 (57.4%) | 0.874 |
| Days with disease before hospitalization | 7.79 (5.48) | 8 (5.53) | 6.55 (5.04) | 0.078 |
| qSOFA | 0.433 (0.83) | 0.331 (0.78) | 1.08 (0.859) | <0.001 |
| Results chest x-ray | | | | |
| Both lungs affected | 217 (68.2%) | 183 (67.5%) | 34 (72.3%) | 0.612 |
| One lung affected | 59 (18.6%) | 49 (18.1%) | 10 (21.3%) | 0.684 |
| None | 42 (13.2%) | 39 (14.4%) | 3 (6.38%) | 0.165 |
| NIH Clinical Presentation | | | | |
| Mild | 0 (0%) | 0 (0%) | 0 (0%) | |
| Moderate | 19 (6%) | 19 (6%) | 0 (0%) | |
| Severe & Critical | 299 (94%) | 252 (92,9%) | 47 (100%) | |
| Fever | 241 (76%) | 210 (77.5%) | 31 (67.4%) | 0.14 |
| Dyspnoea | 164 (51.6%) | 136 (50.2%) | 28 (59.6%) | 0.27 |
| Cough | 200 (63.3%) | 177 (65.6%) | 23 (50%) | 0.048 |
| Asthenia | 158 (49.8%) | 130 (48.1%) | 28 (59.6%) | 0.158 |
| Anosmia | 18 (5.66%) | 17 (6.27%) | 1 (2.13%) | 0.49 |
| Ageusia | 22 (6.92%) | 20 (7.38%) | 2 (4.26%) | 0.754 |
| Obesity | 48 (15.2%) | 40 (14.9%) | 8 (17%) | 0.664 |
| Smoking | | | | |
| Ex-smoker | 20 (6.29%) | 17 (6.27%) | 3 (6.38%) | 1 |
| Yes | 39 (12.3%) | 30 (11.1%) | 9 (19.1%) | 0.146 |
| COPD | 24 (7.55%) | 17 (6.27%) | 7 (14.9%) | 0.065 |
| Asthma | 26 (8.18%) | 23 (8.49%) | 3 (6.38%) | 0.779 |
| Hypertension | 164 (51.6%) | 141 (52%) | 23 (48.9%) | 0.753 |
| Chronic Heart Disease | 28 (8.81%) | 22 (8.12%) | 6 (12.8%) | 0.275 |
| Atrial fibrillation | 36 (11.3%) | 28 (10.3%) | 8 (17%) | 0.21 |
| Immunosuppression | 13 (3.77%) | 8 (2.95%) | 4 (8.51%) | 0.084 |
| Tumour | 35 (11%) | 26 (9.59%) | 9 (19.1%) | 0.073 |
| ACE inhibitors / ARBs | 131 (41.2%) | 115 (42.4%) | 16 (34%) | 0.336 |
| Pre-hospitalization corticosteroids* | 19 (5.97%) | 14 (5.17%) | 5 (10.6%) | 0.175 |
| Diabetes | 75 (23.6%) | 61 (22.5%) | 14 (29.8%) | 0.27 |
| Vitamin D Levels | 17.6 (33.7) | 18.1 (35.3) | 12.5 (10.9) | 0.173 |
| SARS-CoV-2 PCR Positive | 244 (76.7%) | 204 (75.3%) | 40 (85.1%) | 0.19 |
| SARS-CoV-2 Serology Positive | 43 (13.5%) | 40 (14.8%) | 3 (6.38%) | 0.165 |

Numerical variables are presented as mean (standard deviation). Categorical variables are presented as total number with percentages. P values were computed with a Student t-test (numerical variables) or Fisher's exact test (categorical variables). qSOFA (quick Sequential Organ Failure Assessment). See Methods for a full description of all of these variables. COPD (Chronic Obstructive Pulmonary Disease). ACE inhibitors / ARBs, angiotensin-converting enzyme inhibitors / angiotensin receptor blockers.

*Pre-hospitalization corticosteroids refers to whether the patient was under any oral or inhaled corticosteroid therapy at the time of hospitalisation.

descent. Hypertension and diabetes mellitus were present in a 51.6% and 23.6% of the patients respectively. Other comorbid conditions were infrequent (less than 10%) and did not show any statistically significant differences between survivors and non-survivors (Table 1). As previously reported [17, 18], vitamin D levels were significantly different between survivors and non-survivors (p = 0.025). None of the pre-hospitalisation therapies (i.e. corticosteroids and ACE inhibitors / ARBs) showed statistically significant differences between survivors and non-survivors.

## Study end point

We aimed to investigate which factors and interventions were associated with an increased survival by multivariate Cox regression analysis. Among the 318 patients included in the study, 47 died (14.8%). Table 2 shows the full list of therapeutic interventions and oxygen requirements of the patients.

Multivariate Cox regression controlling for clinical covariates as well as all the treatments the patients received (Fig 1, see Methods), revealed a statistically significant increased death risk with age (HR 1.05 [95% CI 1.01–1.09]; P = 0.009) and high volumes of oxygen requirements

**Table 2. Administered treatments and therapeutic needs.**

| | Total (n = 318) | Survivors (n = 271) | Non-survivors (n = 47) | P value |
|---|---|---|---|---|
| Need for oxygen supplementation | 259 (81.4%) | 215 (79.3%) | 44 (93.6%) | 0.024 |
| Needed high oxygen volume | 74 (23.9%) | 38 (14.4%) | 36 (78.3%) | <0.001 |
| Mechanical Assisted Ventilation (non-invasive) | | | | |
| CPAP | 3 (0.943%) | 2 (0.743%) | 1 (2.22%) | 0.372 |
| High flux Oxygen | 27 (8.49%) | 18 (6.69%) | 9 (20%) | 0.007 |
| Entered Intensive Care | 37 (11.8%) | 24 (8.96%) | 13 (28.3%) | 0.001 |
| Mechanical Assisted Ventilation (invasive) | 25 (7.91%) | 12 (4.44%) | 13 (28.3%) | <0.001 |
| Hydroxychloroquine | 297 (93.4%) | 257 (94.8%) | 40 (85.1%) | 0.022 |
| Azithromycin | 281 (88.6%) | 244 (90.4%) | 37 (78.7%) | 0.042 |
| Lopinavir/Ritonavir | 209 (65.7%) | 180 (66.4%) | 29 (61.7%) | 0.618 |
| Interferon | 37 (11.7%) | 27 (10%) | 10 (21.3%) | 0.045 |
| High Dose Corticosteroids PT | 64 (20.1%) | 60 (22.1%) | 4 (8.51%) | 0.031 |
| Low Dose Corticosteroids | 68 (21.4%) | 57 (21%) | 11 (23.4%) | 0.702 |
| Tozilizumab | 24 (7.59%) | 17 (6.32%) | 7 (14.9%) | 0.066 |
| Immunoglobulins | 3 (0.943%) | 3 (1.11%) | 0 (0%) | 1 |
| Anakinra | 2 (0.629%) | 2 (0.738%) | 0 (0%) | 1 |
| Vitamin D Supplementation | 37 (11.6%) | 36 (13.3%) | 1 (2.13%) | 0.025 |
| Anticoagulants: prophylactic dose | 233 (73.3%) | 200 (74.6%) | 33 (70.2%) | 0.589 |
| Anticoagulants: intermediate dose | 24 (7.55%) | 22 (8.18%) | 2 (4.26%) | 0.551 |
| Anticoagulants: full dose | | | | |
| DOACs | 5 (1.57%) | 5 (1.86%) | 0 (0%) | 1 |
| LMWH | 83 (26.1%) | 61 (22.7%) | 22 (46.8%) | 0.001 |

CPAP, continuous positive airway pressure. High Dose Corticosteroids PT, high dose corticosteroids pulse therapy. DOACs, direct-acting oral anticoagulants. LMWH, low-molecular-weight-heparin. Numerical variables are presented as mean (standard deviation). Categorical variables are presented as total number with percentages. P values were computed with a Student t-test (numerical variables) or Fisher's exact test (categorical variables).

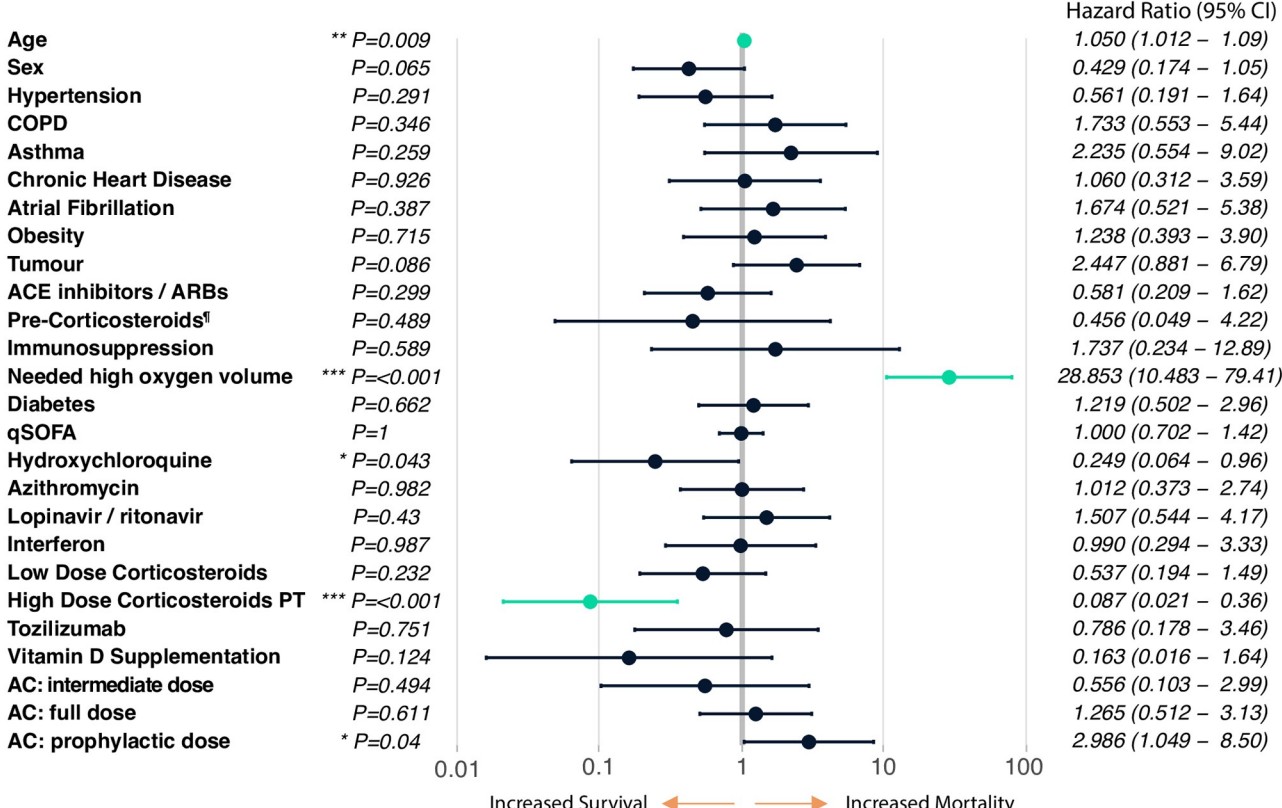

**Fig 1. Multivariate Cox regression analysis, n = 318.** Pre-Corticosteroids¶ refers to whether the patient was under any oral or inhaled corticosteroid therapy upon entering the hospital. COPD (Chronic Obstructive Pulmonary Disease). ACE inhibitors / ARBs, angiotensin-converting enzyme inhibitors / angiotensin receptor blockers. qSOFA (quick Sequential Organ Failure Assessment). High Dose Corticosteroids PT, high-dose corticosteroids pulse therapy. AC, anticoagulation therapy. See Methods for a full description of all of these variables. Highlighted in green are the interventions with p-value < 0.01 (**/***).

(>10L, HR 28.85 [95% CI 10.48–79.41]; P < 0.001). Prophylactic anticoagulation showed a less statistically significant detrimental effect (HR 2.99 [95% CI 1.05–8.50], P = 0.04). No other interventions showed a statistically significant increase in mortality rate. High dose corticosteroid pulse therapy showed a statistically significant reduction in mortality (HR = 0.087 [95% CI 0.021–0.36]; P < 0.001). Hydroxychloroquine was the only other intervention that showed some statistical evidence for a reduction in mortality, although only at the P < 0.05 level (HR = 0.249 [95% CI 0.064–0.96]; P = 0.043).

## Time course analysis of laboratory markers

We carried out a time course analysis of forty-five different laboratory markers over the first month since disease onset distinguishing between COVID-19 survivors and non-survivors, n = 318 (see Methods and S1 Table). Statistically significant levels (FDR < 0.05) were found for thirty markers (see S1 Table). Among these, we highlight time course differences in the following pro-inflammatory markers: IL-6, Ferritin, Lactate dehydrogenase (LDH), D-dimer and C-reactive protein (CRP, Fig 2a). Due to the utility in clinical decision making, we also highlight overall time differences in platelets, total neutrophils, troponin T, total lymphocytes, procalcitonin, glomerular filtration rate (GFR) and triglycerides (Fig 2b).

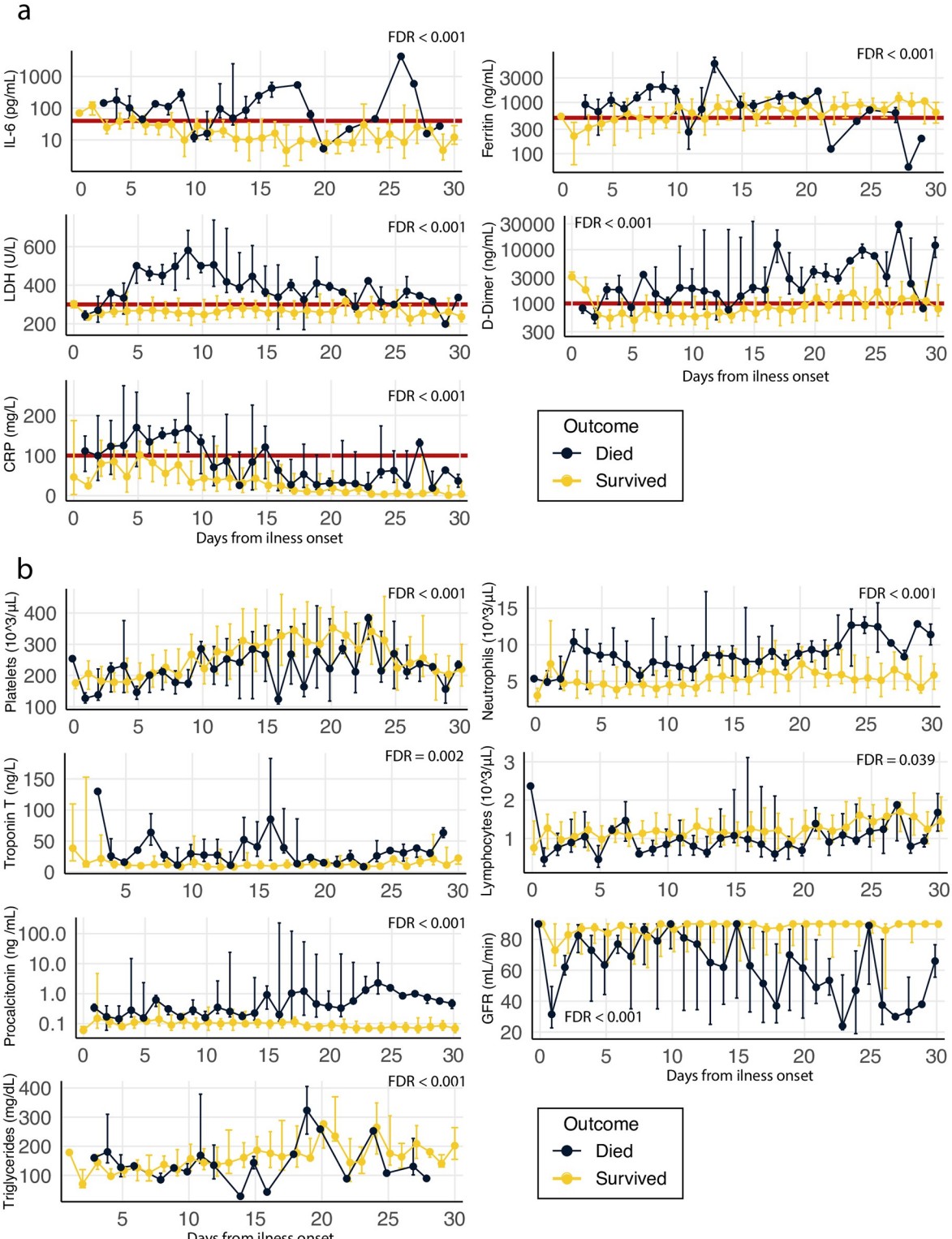

**Fig 2. Time course analysis of survivors/non-survivors all laboratory tests carried out during the first month from COVID-19 illness onset.** Median and interquartile range are presented in each timepoint. The data is showed in a linear scale except in the case of IL-6, ferritin, D-dimer and procalcitonin, which are shown in a Log10 scale (see S1 Table for the measurement unit of all markers and rest of the time course analyses). The false discovery rate (FDR) of the overall time-differences is shown for each marker (see the results of all the tests carried out in S1 Table). **a**. Inflammatory markers: IL-6, Ferritin, Lactate dehydrogenase (LDH), D-dimer and C-reactive protein (CRP).

Marked with a red line are suggestive cut-offs that could potentially be used to select patients at risk of developing a hyper-inflammatory response IL-6 > = 40 pg/ml, C-reactive protein > = 100 mg/L, D-dimer > = 1000 ng/ml, ferritin > = 500 ng/ml and lactate dehydrogenase > = 300 U/L). **b**. Other clinically relevant markers: platelets, total neutrophils, troponin T, total lymphocytes, procalcitonin, glomerular filtration rate (GFR) and triglycerides.

By following this time-course analysis of pro-inflammatory markers between survivors and non-survivors, we could propose an initial COVID-19 specific criteria to diagnose the development of COVID-19 hyper-inflammatory response as following: patients with IL-6 > = 40 pg/ml, and/or two of the following: C-reactive protein > = 100 mg/L, D-dimer > = 1000 ng/ml, ferritin > = 500 ng/ml and lactate dehydrogenase > = 300 U/L (Fig 2a, marked with a red line).

## Discussion

In this study, we show that in patients infected with SARS-CoV-2, the use of pulses of corticosteroids may increase survival. Several studies have reported that the use of corticosteroids may not be beneficial in illnesses caused by other coronaviruses (such as SARS-CoV-1 and MERS-CoV) [8, 10]. On the contrary, even without published scientific evidence [19], other authors [20–23] recommended their use to stop the hyper-inflammatory response following the observed hyper-inflammatory phase and its similarities to the inflammatory phases observed in other diseases such as hemophagocytic syndrome or macrophage activation syndrome [24]. Callejas *et al* [14] recommended the use of high dose corticosteroids pulses to reduce the need for assisted ventilation and death. It is worth nothing that in this study we define high dose corticosteroids pulses as doses of at least 125 mg of methylprednisolone or dexamethasone equivalent. Previous studies in COVID-19 patients did not found clinical differences between methylprednisolone doses above 125 mg [14, 25, 26]. We also tested for low corticosteroid doses and did not find a statistically significant difference in outcome. However, they were given for at most five days, so it cannot be ruled out that they may be effective with a longer course of treatment.

To determine which patients were likely to develop hyper-inflammatory response, and therefore decide which patients should be given HDCPT, we mostly followed the criteria previously suggested from empirical observations and the guidelines used for the macrophage activation syndrome (IL-6 > = 40 pg/ml, and/or two of the following: D-dimer > = 1000 ng/ml, ferritin > = 300 ng/ml and triglycerides > = 300 mg/dL) [14]. However, even though there are similarities between the inflammatory reaction observed in this disease and COVID-19, our time-course analysis of pro-inflammatory markers between survivors and non-survivors showed some marked differences, from which we could derive specific criteria to diagnose the development of COVID-19 hyper-inflammatory response. Referring back to the criteria suggested by Callejas *et al* [14], we would keep the same cut-offs for IL-6 (IL-6 > = 40 pg/ml) and D-dimer (D-dimer > = 1000 ng/ml) and would raise the limit of ferritin to 500ng/ml, since both survivors and non-survivors presented average ferritin levels over 300ng/ml. We did not observe clear differences that could distinguish between survivors and non-survivors for triglyceride levels, because of this, we removed this marker and instead we suggest including C-reactive protein and lactate dehydrogenase as indirect markers of inflammation at > = 100 mg/L and > = 300 U/L respectively.

The patients included in this study were also under other drugs including the anti-inflammatory drug tozilizumab (used in 15 patients with IL-6 levels higher than 40 pg/ml). Even though we observed a survival rate of 73.1% in the patients who took tozilizumab, the overall results were not statistically significant (which could be due to the low sample size). However, the observed increased trend for survival is in line with what was published by Lou

*et al* [12] and Campins *et al* [27]. In their study, they found an increased survival rate for tozilizumab in patients with early intervention and several doses.

The role of hydroxycholoquine in COVID-19 remains controversial. In our study we only found a marginal association between the use of hydroxycholoquine and an increase in survival rate. Although this study was not designed to assess the role of hydroxycholoquine in survival rates, this marginal association could be in line with what had been reported in previous studies in which they found that hydroxychloroquine was effective at inhibiting SARS-CoV-2 in vitro [28]. On the contrary, randomized clinical trials such as RECOVERY [29] and the one carried out by Cavalcanti *et al* [30] did not found an increase in survival rates in COVID-19 patients. However, the clinical study carried out by Cavalcanti only looked at mild and/or moderate COVID-19 patients and the RECOVERY study did not study severe hospitalized patients. To date, there is not enough evidence to suggest that hydroxycholoquine is effective at increasing the survival of COVID-19 patients and more studies are needed to dissect the effects of hydroxycholoquine in COVID-19. Regarding other antiviral therapies, even though some potential beneficial effects have been described for azithromycin [31], lopinavir/ritonavir [32] and interferon [33], we did not find any statistically significant increase in survival with any of these, or a combination of them.

Extensive clotting has also been observed in COVID-19 [34–36], which may evidence the need antithrombotic medication therapy in all the patients with high levels of D-dimer or hints of initial disseminated intravascular coagulation [37]. Nonetheless, this is still a matter of debate and recent studies have also questioned the need for a full anticoagulation dose unless there are further clinical evidence supporting this need. In our study, even though we found a marginally statistically significant association between prophylactic anticoagulation and mortality, our analyses were not designed to ask this question but instead looking at interventions affecting overall inflammation associated with death. Therefore, we do not conclude that a prophylactic dose increases mortality risk. To answer this question, we would have to specifically stratify the patients based on D-dimer levels. Further studies should assess the different types of anticoagulation and their association with disease outcomes.

Other studies have shown that corticosteroid therapy does not affect virus clearance time [38]. Unfortunately, in our study we did not carry out a follow up quantify viral clearance. It would be interesting to assess whether HDCPT has an impact on virus clearance times and we hope that future studies can shed light on this topic.

This study has some major limitations including that all the patients were from a single centre and a single ethnic group. Moreover, even though we have carried out multivariate analyses to account for any possible confounding effects, various imbalances may possibly exist between patient groups. This study is not a randomized clinical trial and therefore causality cannot be derived. However, there is a promising effect for high dose corticosteroid pulse therapy for improving severe/critical COVID-19 disease progression and increasing the survival rates of patients at risk of developing a hyper-inflammatory response. We also suggest some initial criteria using pro-inflammatory markers to diagnose these patients. Future multicentre randomized clinical trials should be carried out to confirm the efficacy of high pulse corticosteroids pulses therapy to increase the survival rates of COVID-19.

## Supporting information

**S1 Table. All forty-five laboratory makers tested.** P-value and false discovery rate (FDR) of 30-day time-course analysis between survivors and non-survivors are included. Markers with statistically significant changes are highlighted in grey.
(DOCX)

## Acknowledgments

Dr Jose Luis Callejas-Rubio, Dr Malcolm Perry, Dr Ascensión María Vílchez-Parras, Mr Pablo Sánchez-Moya, Ms Ana Ramírez-Sánchez and Dr Raul Elgueta for the helpful discussions and their help to carry out this project.

## Author Contributions

**Conceptualization:** Miguel Ángel López Ruz.

**Data curation:** Miguel Ángel López Zúñiga, Ana Ocaña-Granados, Francisco Andrés Padilla-Moreno, Alba María Castillo-Fernández, Dionisio Guillamón-Fernández, Carolina Ramírez-Sánchez, María Sanchez-Palop, Justo Martínez-Colmenero, María Amparo Pimentel-Villar, Sara Blázquez-Roselló, José Juan Moreno-Sánchez, María López-Vílchez, Inmaculada Prior-Sánchez, Rosario Jódar-Moreno.

**Formal analysis:** Miguel Ángel López Zúñiga, Aida Moreno-Moral.

**Investigation:** Miguel Ángel López Zúñiga, Ana Ocaña-Granados, Francisco Andrés Padilla-Moreno, Alba María Castillo-Fernández, Justo Martínez-Colmenero, Miguel Ángel López Ruz.

**Methodology:** Miguel Ángel López Zúñiga.

**Resources:** Miguel Ángel López Ruz.

**Validation:** Miguel Ángel López Zúñiga, Miguel Ángel López Ruz.

**Writing – original draft:** Miguel Ángel López Zúñiga, Aida Moreno-Moral.

**Writing – review & editing:** Aida Moreno-Moral, Miguel Ángel López Ruz.

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
