## [Decision Letter · Decision Letter 0]

12 Oct 2020

PONE-D-20-28257

High-Dose Corticosteroid Pulse Therapy increases the survival rate in COVID-19 patients at risk of cytokine storm

PLOS ONE

Dear Dr. López Zúñiga,

Thank you for submitting your manuscript to PLOS ONE. After careful consideration, we feel that it has merit but does not fully meet PLOS ONE’s publication criteria as it currently stands. Therefore, we invite you to submit a revised version of the manuscript that addresses the points raised during the review process.

We look forward to receiving your revised manuscript.

Kind regards,

Wenbin Tan

Academic Editor

PLOS ONE

2. Please provide additional details regarding participant consent. In the ethics statement in the Methods and online submission information, please specify how verbal consent was documented and witnessed.

3. For studies involving humans categorized by race/ethnicity, age, disease/disabilities, religion, sex/gender, sexual orientation, or other socially constructed groupings, authors should:

1) Explicitly describe their methods of categorizing human populations,

2) Define categories in as much detail as the study protocol allows,

3) Justify their choices of definitions and categories,

4) Explain whether (and if so, how) they controlled for confounding variables such as socioeconomic status, nutrition, environmental exposures, or similar factors in their analysis, and

5) Update outmoded terms and potentially stigmatizing labels to more current, acceptable terminology.

Examples: “Caucasian” should be changed to “white” or “of [Western] European descent” (as appropriate); “XXX victims” should be changed to “patients with XXX.

Reviewer #1: A good job. This is a well-designed and conducted prospective observational study. The conclusion is HDCPT can decrease the mortality of COVID 19 patients with signs of cytokine storm.

I have a few minor concerns:

1. Who took the low dose corticosteroid therapy?

2. "16 patients received HDCPT due to their critical clinical status." Can you explain it in detail?

3. Last, did all the patients have laboratory tests every day until they discharged or died? No matter they stayed in ICU or not? Please provide details.

Reviewer #2: For this study the authors collected data in an observational and prospective fashion for confirmed or suspected COVID-19 cases. They computed and followed up various laboratory markers to distinguish survivors and non-survivors, thus to describe a group of patients with poor prognosis whom may require high-dose pulse steroid therapy (>1.5mg/kg/d methylprednisolone or equivalent dose of dexamethasone) for 2-5 days. Several factors were compared retrospectively between survivors and non-survivors.

This is well written manuscript however suffers from the fact that it is not randomized controlled trial. As the authors also admitted there many confounding factors which may have affected their results. There have been already several randomized controlled trials including RECOVERY trial and and at least one meta-analysis (.doi:10.1001/jama.2020.17023) looking for the effect of steroids in critically ill COVID-19 patients. RECOVERY trial tested a lower dose of dexamethasone (i.e. 6 mg/d) than the current study albeit longer duration. Thus, this study may have had a conclusion saying that high doses of steroids (i.e. 1,5 mg/kg/d) for short courses may have an impact on mortality. However, the design of this study would not allow such conclusions.

My further comments include the following:

1) The authors refer a "cytokine storm" throughout the text which has been severely criticised in the literature (doi:10.1001/jamainternmed.2020.3313). Perhaps "hyper-inflammatory response" is a better term which has also been used alternatively.

2) The authors defined pre-set criteria for high dose steroid use, but less than one-third of patients were given the drug. In addition, 16 patients without these criteria received it. This is a clear bias. Since the study is not randomized and controlled, various imbalances may possibly exist in groups receiving and non-receiving high dose steroids.

3) Defining high-dose of steroids is also arbitrary. In general high dose pulse steroid may usually refer dose equalling 250 mg or higher methylprednisolone given with short duration of infusion.

4) The duration of steroid therapy is defined as 2-5 days which lacks standardization and the effects of therapy may differ between those who received 2 days vs 5 days.

---

## [Author Response · Author response to Decision Letter 0]

21 Oct 2020

Response to reviewers: 

Dear reviewers,

First of all, thank you for the time dedicated and the proposals you have made on the “High-Dose Corticosteroid Pulse Therapy increases the survival rate in COVID-19 patients at risk of hyper-inflammatory response” by Lopez-Zuñiga et al. Below we respond to your suggestions.

Yours sincerely,

The Autors.

Reviewer #1: A good job. This is a well-designed and conducted prospective observational study. The conclusion is HDCPT can decrease the mortality of COVID 19 patients with signs of cytokine storm.

I have a few minor concerns:

 1. Who took the low dose corticosteroid therapy? 

We thank the reviewer pointing out this missing information, we have now added this to the methods as follows:

“Low dose corticosteroid therapy was administered to patients who had had a bronchospasm, following standard clinical guidelines.” 

2. "16 patients received HDCPT due to their critical clinical status." Can you explain it in detail?

We thank the reviewer for requesting this clarification, these details have been added to the methods section as follows:

“These 16 patients received high dose corticosteroids pulse therapy because they developed severe respiratory failure and did not respond to the standard COVID-19 clinical practice at that time, including pharmacological (hydroxychloroquine, azithromycin and lopinavir-ritonavir) and physical interventions (i.e. postural changes).”

3. Last, did all the patients have laboratory tests every day until they discharged or died? No matter they stayed in ICU or not? Please provide details.

These details have now been added to the text:

“All the patients who were in ICU had laboratory tests every 24h. Patients outside of ICU had laboratory tests every 48h unless showing worsening symptoms (in these cases they were tested every 24h).” 

Reviewer #2: For this study the authors collected data in an observational and prospective fashion for confirmed or suspected COVID-19 cases. They computed and followed up various laboratory markers to distinguish survivors and non-survivors, thus to describe a group of patients with poor prognosis whom may require high-dose pulse steroid therapy (>1.5mg/kg/d methylprednisolone or equivalent dose of dexamethasone) for 2-5 days. Several factors were compared retrospectively between survivors and non-survivors.

This is well written manuscript however suffers from the fact that it is not randomized controlled trial. As the authors also admitted there many confounding factors which may have affected their results. There have been already several randomized controlled trials including RECOVERY trial and and at least one meta-analysis (.doi:10.1001/jama.2020.17023) looking for the effect of steroids in critically ill COVID-19 patients. RECOVERY trial tested a lower dose of dexamethasone (i.e. 6 mg/d) than the current study albeit longer duration. Thus, this study may have had a conclusion saying that high doses of steroids (i.e. 1,5 mg/kg/d) for short courses may have an impact on mortality. However, the design of this study would not allow such conclusions.

My further comments include the following:

 1) The authors refer a "cytokine storm" throughout the text which has been severely criticised in the literature (doi:10.1001/jamainternmed.2020.3313). Perhaps "hyper-inflammatory response" is a better term which has also been used alternatively.

We thank the reviewer for his/her suggestion. We have now replaced all the instances in the text. 

 2) The authors defined pre-set criteria for high dose steroid use, but less than one-third of patients were given the drug. In addition, 16 patients without these criteria received it. This is a clear bias. Since the study is not randomized and controlled, various imbalances may possibly exist in groups receiving and non-receiving high dose steroids.

We thank the reviewer for querying further into this. We agree with the reviewer and acknowledge the limitations of the study design. Overall, we have not observed any major biases in any of the variables measured and our analyses are carried out using multivariate approaches. However, we have amended the discussion and abstract to reflect these limitations, and they are further acknowledged in our conclusions.

At the time this study was performed, some experts hypothesized the presence of a hyperinflammatory syndrome, which could point to a beneficial role for corticosteroids in treating COVID-19. However, the World Health Organisation warned against the use of corticosteroids in patients with SARS-Cov-2. Because of this, some doctors did not want to treat their patients without scientific evidence and therefore some patients were not treated with corticosteroids, despite having similar clinical profiles. In addition, 16 patients were given corticosteroids due to the critical status despite not meeting the pre-defined criteria. Our pre-defined criteria was set following the guidelines used for another other pro-inflammatory syndrome (macrophage activation syndrome) and our analysis of pro-inflammatory markers suggests that this criteria could be re-defined for COVID-19 patients. 

 3) Defining high-dose of steroids is also arbitrary. In general high dose pulse steroid may usually refer dose equaling 250 mg or higher methylprednisolone given with short duration of infusion.

We thank the reviewer for querying the dose. To set the high-dose of corticosteroids we followed guidelines from studies in other autoimmune diseases in which they found that, above methylprednisolone doses of 125 mg there were no clinical differences. Recent studies in COVID-19 patients have also corroborated this (Callejas et al 10.1016/j.medcli.2020.04.018 and Ruiz-Irastorza et al 10.1371/journal.pone.0239401). 

We have added this to the discussion as follows: 

“In this study we define high dose corticosteroids pulses as doses of at least 125 mg of methylprednisolone or dexamethasone equivalent. Previous studies in COVID-19 patients did not found clinical differences between methylprednisolone doses above 125 mg [Callejas et al and Ruiz-Irastorza et al]”

4) The duration of steroid therapy is defined as 2-5 days which lacks standardization and the effects of therapy may differ between those who received 2 days vs 5 days.

We thank the reviewer for querying this. The treatment duration has now been clarified in the text as follows:

“The standard high dose corticosteroid pulse therapy duration was 3 days. In some patients who did not improve after 3 days, the treatment was extended to 5 days. In two patients, the treatment was shortened to 2 days due to the significant recovery observed.”

---

## [Decision Letter · Decision Letter 1]

30 Oct 2020

PONE-D-20-28257R1

High-Dose Corticosteroid Pulse Therapy increases the survival rate in COVID-19 patients at risk of hyper-inflammatory response

PLOS ONE

Dear Dr. López Zúñiga,

Thank you for submitting your manuscript to PLOS ONE. After careful consideration, we feel that it has merit but does not fully meet PLOS ONE’s publication criteria as it currently stands. Therefore, we invite you to submit a revised version of the manuscript that addresses the points raised during the review process.

We look forward to receiving your revised manuscript.

Kind regards,

Wenbin Tan

Academic Editor

PLOS ONE

Reviewer #3: Authors observed the increased survival in COVID-19 patients with HDCPT and proposed a potential COVID-19 specific criteria to diagnose the development of COVID-19 cytokine storms. It is a very interesting study, for which could clinically direct doctors to use HDCPT for the treatment of COVID-19 patients. However, there still have several questions need to address.

1. The current results have shown that HDCPT is effective. it is better to divide the patients into mild/moderate,  severe and critical categories and observe survival rates after HDCPT based on the guidelines from NIH or Europe (such as https://www.covid19treatmentguidelines.nih.gov/overview/clinical-presentation/).

2. Virus clearance time should be addressed the study to value the adverse effects with HDCPT patients.

3. In figure 2, the laboratory tests also need to carried out in severity categories, which will make the differences more significant.

4. Several studies had reported that hydroxychloroquine have no benefit on COVID-19 patients, which seems controversial with the result. Please discuss this point in the discussion section .

---

## [Author Response · Author response to Decision Letter 1]

16 Nov 2020

Dear reviewers,

First of all, thank you for the time dedicated and the proposals you have made on the “High-Dose Corticosteroid Pulse Therapy increases the survival rate in COVID-19 patients at risk of hyper-inflammatory response” by Lopez-Zuñiga et al. Below we respond to your suggestions.

Yours sincerely,

The Authors.

Authors observed the increased survival in COVID-19 patients with HDCPT and proposed a potential COVID-19 specific criteria to diagnose the development of COVID-19 cytokine storms. It is a very interesting study, for which could clinically direct doctors to use HDCPT for the treatment of COVID-19 patients. However, there still have several questions need to address.

1. The current results have shown that HDCPT is effective. it is better to divide the patients into mild/moderate, severe and critical categories and observe survival rates after HDCPT based on the guidelines from NIH or Europe (such as https://www.covid19treatmentguidelines.nih.gov/overview/clinical-presentation/).

We thank the reviewer for his/her suggestion to inspect the results by disease category. According to the NIH guidelines, out of the 318 patients included in our study, 20 patients were moderate (6.3%). The rest of the patients were all severe or critical (n=238, 93.4%). Unfortunately, we cannot distinguish between severe and critical patients, since we do not have data for respiratory or multi organ failure, and not all critical patients were admitted to intensive care due to overcrowding at the height of the pandemic. We have now included these details in the revised manuscript. 

In addition, we have recomputed our multivariate analysis after removing moderate patients, the results are very similar:

Analysis without moderate patients (n=298) 

Analysis with all patients (presented in the manuscript, n=318)

2. Virus clearance time should be addressed the study to value the adverse effects with HDCPT patients.

We agree with the reviewer that it would be very interesting to examine the effects of HDCPT on virus clearance time. Unfortunately, during the time of the study we did not collect any information to tackle this issue. In the revised version of the manuscript have added virus clearance time to the discussion as one key factor to cover in future studies as following: 

Other studies have shown that corticosteroid therapy does not affect virus clearance time [10.1038/s41598-020-70387-2]. Unfortunately, in our study we did not carry out a follow up quantify viral clearance. It would be interesting to assess whether HDCPT has an impact on virus clearance times and we hope that future studies can shed light on this topic. 

3. In figure 2, the laboratory tests also need to carried out in severity categories, which will make the differences more significant

Similarly to the previous point, we have reanalysed the data removing all the moderate patients. The results are quite similar and in some cases the results are slightly more significant. 

Analysis without moderate patients (n=298)

Analysis with all patients (presented in the manuscript, n=318)

4. Several studies had reported that hydroxychloroquine have no benefit on COVID-19 patients, which seems controversial with the result. Please discuss this point in the discussion section.

We thank the reviewer for suggesting a further discussion on the effect of hydroxycholoquine as it is a controversial topic in this COVID-19. We have now added this to the discussion as follows:

The role of hydroxycholoquine in COVID-19 remains controversial. In our study we only found a marginal association between the use of hydroxycholoquine and an increase in survival rate. Although this study was not designed to assess the role of hydroxycholoquine in survival rates, this marginal association could be in line with what had been reported in previous studies in which they found that hydroxychloroquine was effective at inhibiting SARS-CoV-2 in vitro [ ]. On the contrary, randomized clinical trials such as RECOVERY [ ] and the one carried out by Cavalcanti et al [ ] did not found an increase in survival rates in COVID-19 patients. However, the clinical study carried out by Cavalcanti only looked at mild and/or moderate COVID-19 patients and the RECOVERY study did not study severe hospitalized patients. To date, there is not enough evidence to suggest that hydroxycholoquine is effective at increasing the survival of COVID-19 patients and more studies are needed to dissect the effects of hydroxycholoquine in COVID-19.

---

## [Decision Letter · Decision Letter 2]

2 Dec 2020

High-Dose Corticosteroid Pulse Therapy increases the survival rate in COVID-19 patients at risk of hyper-inflammatory response

PONE-D-20-28257R2

Dear Dr. López Zúñiga,

We’re pleased to inform you that your manuscript has been judged scientifically suitable for publication and will be formally accepted for publication once it meets all outstanding technical requirements.

Kind regards,

Wenbin Tan

Academic Editor

PLOS ONE

Review Comments to the Author

Reviewer #3: The authors have made revisions according to reviewers' comments and it look‘s better in the revised manuscript.

---

## [Editor Report · Acceptance letter]

20 Jan 2021

PONE-D-20-28257R2 

High-Dose Corticosteroid Pulse Therapy increases the survival rate in COVID-19 patients at risk of hyper-inflammatory response 

Dear Dr. López-Zúñiga:

I'm pleased to inform you that your manuscript has been deemed suitable for publication in PLOS ONE. Congratulations! Your manuscript is now with our production department. 

Kind regards, 

on behalf of

Dr. Wenbin Tan 

Academic Editor

PLOS ONE